# Effect of Ru on Deformation Mechanism and Microstructure Evolution of Single-Crystal Superalloys under Medium-Temperature and High-Stress Creep

**DOI:** 10.3390/ma16072732

**Published:** 2023-03-29

**Authors:** Stephen Okhiai Emokpaire, Nan Wang, Jide Liu, Chongwei Zhu, Xinguang Wang, Jinguo Li, Yizhou Zhou

**Affiliations:** 1Shi-Changxu Innovation Center for Advanced Materials, Institute of Metal Research, Chinese Academy of Sciences, Shenyang 110016, China; stephen17b@imr.ac.cn (S.O.E.); 2290049@stu.neu.edu.cn (N.W.); cwzhu22b@imr.ac.cn (C.Z.); xgwang11b@imr.ac.cn (X.W.); jgli@imr.ac.cn (J.L.); yzzhou@imr.ac.cn (Y.Z.); 2School of Materials Science and Engineering, University of Science and Technology of China, Shenyang 110016, China; 3The State Key Laboratory of Rolling and Automation, Northeastern University, Shenyang 110016, China

**Keywords:** creep deformation, deformation mechanism, γ′ phase, single-crystal superalloys

## Abstract

In this work, the effect of the Ru element on the γ′-phase evolution and deformation mechanism in the fourth-generation Ni-based single-crystal superalloy was investigated. Results show that the Ru element alters the distribution coefficient of other elements in the alloy to produce reverse partitioning behavior, which leads to a difference in microstructure between 0Ru and 3Ru. The addition of Ru triggered the incubation period before the beginning of the primary creep stage, which depends on the creep temperature and stress during creep deformation. TEM results revealed that Ru addition inhibits the slip system {111}<112> at medium-temperature (760–1050 °C) and high-stress (270–810 MPa) creep, which brings a considerably low creep rate and high creep life to the Ru-containing alloy.

## 1. Introduction

Ni-based single-crystal superalloys have superior mechanical properties and corrosion resistance at elevated temperatures, which makes them the most suitable materials for the manufacture of turbine blades in aero engines. Single-crystal superalloy can withstand large axial tensile force along the direction [001] because it eliminates the weak area of grain boundary, and its strength is much higher than polycrystalline and columnar crystal superalloy, so it acts as an irreplaceable category of nickel-based superalloy. The temperature capability of the turbine blade has increased significantly over the past several decades. Some of the advances have been achieved by improving the content of refractory alloying elements [1,2,3,4].

Ni-based single-crystal superalloys have complexity in alloying elements such as Al, Co, Ti, Ta, Cr, Mo, W, and Hf, which serve as the first generation of the alloy. The second and third generations contain 2–6% Re, while the fourth generation contains Ru at different weight percentages. It is widely known that Co, Cr, Ir, Mo, Re, and Ru dissolve into the γ matrix phase of the alloy while Al, Ti, Ta, Nb, Pt, and Hf partition into the γ′ phase, which serves as the precipitated phase in the γ-phase matrix [5,6].

With regards to the partition coefficient of the γ-phase matrix, rhenium tends to have the highest partition coefficient in γ phase as compared to other alloying elements in the Ni-based single-crystal superalloy. The addition of Re to Ni-based single-crystal superalloys brought about an increase in the creep properties of the alloy, which is a result of the increase in the lattice parameter of the γ phase, the increase in the solidus temperature, and the reduction in the coefficient of diffusion [7,8,9]. However, the addition of Re makes the microstructural homogenization during solution heat treatment more difficult due to its low diffusivity and severe micro-segregation. Additionally, at high temperatures and long-term exposure, there is the formation of topologically close-packed (TCP) phases in Re containing superalloys, which becomes detrimental to the creep properties of the alloy by depletion of important strengthening elements from the matrix, the cracks initiation induced by the stress concentration [10,11]. This deleterious effect of Re necessitated the introduction of Ru in Ni-based single-crystal superalloys. Thus, Ru has become a very significant element in Ni-based single-crystal superalloys. It is known to suppress the formation of the TCP phase and also helps to stabilize the microstructure and phase composition of the alloy [12,13,14,15].

It is well established that Ru-containing Ni-based SC superalloys have better creep properties compared to Ru-free Ni-based SC superalloys, especially for high-temperature and low-stress creep conditions. However, the creep property of Ru containing SC at medium-temperature (760–1050 °C) and high-stress (270–810 MPa) creep conditions is still, to a large extent, controversial, especially with regards to the deformation mechanism and γ′-phase evolution, hence the reason for this research.

This work is aimed at studying the effect of Ru on creep deformation and γ′-phase evolution at medium-temperature, high-stress creep conditions for an experimental Ni-based single-crystal superalloy.

## 2. Experimental Procedures

Two experimental Ni-based single-crystal superalloys containing 0 wt.% Ru and 3 wt.% Ru (0Ru and 3Ru represent these two alloys) were prepared by directional solidification to form single-crystal bars along the [001] direction from the polycrystalline ingot in a vacuum induction furnace using the Bridgeman method. All other elements are the same except for the 3Ru having a different balance of Ni, as shown in Table 1. Electron backscatter diffraction (EBSD) was performed on the two experimental Ni-based single-crystal superalloys to ascertain the crystal orientation and was observed to be within 10° deviation for all the cylindrical bars. A two-step solution treatment and aging were performed on both experimental alloy samples as follows: 1315 °C, 16 h + 1325 °C, 16 h + (air cooling) → 1150 °C, 4 h + air cooling → 870 °C, 24 h. The heat treatment process is shown in Figure 1. After heat treatment, parts of the specimens were prepared for scanning electron microscopy (SEM) using different grades of abrasive paper, polished and etched with a mixed solution of copper sulfate (20 g CuSO_4_ + 100 mL HCl + 5 mL H_2_SO_4_ + 80 mL H_2_O) at room temperature within 3–5 s. The microstructure of both experimental superalloys was observed in a JSM-7100F field emission scanning electron microscope. Cylindrical-shaped creep specimens were prepared from the heat-treated bars having a gauge length of 25 mm and diameter of 5 mm. The creep test was carried out on an AG-250KNE mechanical testing machine at different temperatures and stress conditions: 760 °C/810 MPa, 850 °C/620 MPa, 850 °C/750 MPa, 950 °C/320 MPa, and 1050 °C/270 MPa to rupture. After creep tests, the fracture surfaces of the specimens were cleaned in ethanol using an ultrasonic cleaner. The fracture surface morphology of specimens was then observed in SEM. Additionally, the post-creep specimens were cut along the longitudinal axis of the gauge section, prepared, and the morphologies of the microstructure were observed in SEM. Thin foils with a thickness of 500 µm were cut from crept specimens at about 5 mm away from the fracture surface and perpendicular to the gauge length section for transmission electron microscope (TEM) examination. The thin foils were mechanically ground to approximately 50 µm using abrasive papers with different grade sizes. The foils were electrochemically thinned using a twin-jet polisher in a solution of 10% perchloric acid and 90% methanol at −22 °C ± 3 and a voltage of 20 V. The thinned foils were then observed in a JEM-2100 TEM operating at 200 KV. All TEM observations were carried out under two-beam bright field imaging. EPMA was used to measure the composition and to obtain the distribution ratio of each alloy element of γ and γ′ phases in the alloy, respectively. The size of γ and γ′ phases in the initial heat treatment structure of nickel-based single-crystal alloys is small. In order to accurately measure the composition of γ and γ′ phases, the samples of both alloys after complete heat treatment were subjected to special coarsening heat treatment at 1300 °C/1 h + (1300–1150 °C)/10 h, followed by air cooling to obtain coarse γ and γ′ phases so that higher resolution can be obtained at lower magnification from an electron probe microanalyzer (EPMA).

## 3. Results and Discussion

### 3.1. Initial Microstructures of Experimental Ni-Based Single-Crystal Superalloys

Figure 1 shows the as-cast microstructures of both experimental Ni-based single-crystal superalloys. It can be observed that both alloys have a typical dendritic structure in the [001] direction, in which the darker color is the dendritic stem region, and the bright area is the final solidified interdendritic γ/γ′ eutectic structure. The addition of Ru, as shown in Figure 2b, indicates that the eutectic structure of the alloy increased, which portrays that Ru alters the distribution coefficient of other elements in the alloy between the dendritic stem and interdendritic region. This is consistent with the study by Liu et al. and Shi et al. [16,17]; however, other studies by Hobbs et al. and Kearsey et al. show otherwise [18,19]. This could be a result of differences in other components of the experimental alloy being investigated; that is, the differences in elemental composition in the alloy may affect the percentage of eutectic content. This means that there is an increase in the volume fraction of the eutectic region with the addition of Ru, while the primary dendrite arm spacing decreases a little.

The heat-treated microstructures of nickel-based superalloys has an important effect on the mechanical properties of the alloys, such as the morphology, size, and volume fraction of the precipitated phase. Figure 2c,d show a typical two-phase microstructure consisting of the γ matrix and γ′ precipitate of the two experimental alloys after full heat treatments. After two-stage solid solution and two-stage aging heat treatments, a regular γ′ strengthening phase precipitate was observed in both alloys, as shown in Figure 1d and Figure 2c. It can be seen from the figure that there is no difference between the interdendritic structure and the dendritic structure of the two alloys, which indicates that the element diffusion of the alloy is relatively effective after two-stage long-term solution heat treatment. However, there are differences in the microstructure of γ′ phases of the 0Ru and 3Ru alloys.

The average size of the γ′ phase of 0Ru is approximately 0.31 μm, while that of the 3Ru alloy is 0.30 μm. The 3Ru alloy had a more uniform precipitated phase as compared to the 0Ru alloy. The standard deviations of the γ′ phase of 0Ru alloy and 3Ru alloy are 0.068 nm and 0.066 nm, respectively. In addition, the volume fraction of the γ′ phase is basically the same (about 69 vol.%), whereas the γ channel width is between 50 and 60 nm. It can be seen from Figure 3 that the addition of Ru reduces the distribution coefficient of Re in the matrix and other γ-strengthening elements. This implies that there is reverse partitioning behavior with the addition of Ru. This reverse partitioning behavior means that Ru promotes the γ-forming elements, such as Re, Mo, and Cr, to partition to the γ phase and the γ′-forming elements, such as Al, Ta, and Ni, to partition to the γ matrix of the alloy. This behavior is consistent with other studies, and it is widely attributed to be the main reason for the suppression of the TCP phase by Ru additions [20,21].

### 3.2. Creep Behavior and Microstructure Evolution of Both Alloys at Different Conditions

#### 3.2.1. Creep Curve

The creep curves of both alloy 0Ru and 3Ru at different conditions of temperature and stress are shown in Figure 4. The creep life for 3Ru at 760 °C/810 MPa condition has a noticeably longer creep time than 0Ru (278 h and 147 h, respectively), especially the steady creep stage, which occupies a large part of the whole creep rupture life as seen in Figure 4a. Alloys 0Ru at this condition followed the typical three creep stages before rupture, i.e., primary creep stage, secondary creep stage, and tertiary creep stage, while there were four stages in alloy 3Ru as shown in the inset of Figure 4a. There was a slight steady creep strain stage in alloy 3Ru before the setting of the typical primary creep stage as compared to 0Ru. This means that there is the absence of an incubation period in the 0Ru alloy. The creep rate of the 0Ru alloy was observed to be very high at 850 °C/620 MPa condition, as indicated in Figure 4b. The creep rupture life of the 3Ru alloy is almost twice that of the 0Ru. It is clearly seen that both alloys exhibited the three creep stages in which both alloys showed longer steady creep rates during the secondary creep stage. However, it is worthy of note that the secondary creep stage of the 3Ru alloy lasted for more than 500 h with an extended period of slowly accelerating tertiary creep, while that of the 0Ru alloy was for about 320 h. The inset shows a 40 h steady creep duration at 850 °C/620 MPa conditions. Figure 4c shows the creep curve for both alloys at 850 °C/750 MPa creep conditions. There is no instantaneous plastic straining upon application of the stress and temperature for 3Ru alloy, as observed in the inset. It is also seen that the experimental alloy 3Ru exhibited a steady-state creep rate for about 4 h before the typical primary creep stage. This means that at the 850 °C/750 MPa creep condition, the 3Ru superalloy has a longer incubation period as compared to the 0Ru alloy before the primary creep stage sets in. The 0Ru alloy shows a higher creep rate. However, both alloys have a shorter creep rupture life compared to the condition of 850 °C/620 MPa.

Figure 4d depicts the creep curves of 0Ru and 3Ru alloys at constant temperatures and stress levels (950 °C/320 MPa). At the same creep circumstances, the variation tendency of the curves of the 0Ru alloy is similar to that of the 3Ru alloy, as illustrated in the inset. The creep life of the alloy increases after the addition of Ru, as shown in the graph. Except for the unusually rapid increase in creep rate prior to the steady state in the 3Ru alloy compared to the 0Ru alloy clearly shown in the inset. However, the creep curves of both experimental alloys at 1050 °C/270 MPa follow similar trends.

#### 3.2.2. Microstructure Evolution after Creep Rupture

Figure 5 exhibits the typical fracture surface morphology of 0Ru and 3Ru experimental superalloys, including microcracks and voids, which is the main cause of creep cracking [22,23]. As observed, the typical fracture mode under this creep condition for both alloys is micro-void aggregation fracture with some obvious necking. There is the presence of microcracks and voids in both alloys. However, the 0Ru alloy tends to have more microcracks and voids, as shown in Figure 5a–f. The propagation of the micro-voids and cracks could influence the creep rupture life of both alloys. This could be a result of Ru helping to inhibit the diffusion rate of other elements, thereby reducing the formation of vacancies [24]. Combining Figure 4, one can conclude that fewer micro-voids and cracks in the 3Ru alloy provide the alloy with a longer creep rupture life under varying temperatures and stress conditions. Additionally, most of the micro-voids could be a result of casting, which originated at the interdendritic region or from the slow conglomeration of a high concentration of vacancies, as seen in Figure 5e [25].

The microstructural changes after creep rupture were studied along the longitudinal section of each specimen to analyze the creep behaviors of both experimental alloys. For the intermediate-temperature, high-stress condition of creep, the microstructures of the Ru-free and Ru-bearing alloys for both 0Ru and 3Ru appear to be identical, as shown in Figure 6. For the 0Ru- and 3Ru-bearing alloys, the precipitates generally remained unaltered and had a cuboidal morphology. However, there is a slight coarsening of the γ′ precipitate and a widening of the γ channel. Combining Figure 3, one can conclude that the differences in microstructure between the 0Ru and 3Ru alloys are negligible as a result of the reverse partitioning behavior, which is not intense with the addition of Ru. There was rafting for the 950 °C/320 MPa and 1050 °C/270 MPa stress conditions but no observable presence of topologically close-packed (TCP) phases for both alloys, as indicated in Figure 7. One of the significant features of 3Ru alloy after creep at 1050 °C/270 MPa is that topology inversion that occurred under this condition has a wider substrate in 0Ru compared to the 3Ru alloy.

#### 3.2.3. Deformation Mechanisms 

There are three main deformation mechanisms that may occur between dislocations and γ′ precipitates in nickel-based superalloy during creep deformation: (1) dislocations shearing into γ′ precipitates either through anti-phase boundaries or stacking faults; (2) slipping and cross-slipping of dislocations in γ matrix between γ′ precipitates; and (3) climbing of dislocations along γ/γ′ interfaces, especially at higher temperatures and low stress [26,27].

The dislocation structures of 0Ru and 3Ru alloy after creep tests at different conditions were obtained by TEM, and the results are shown in Figure 8. The large values for the primary creep strain and strain softening of 0Ru alloy are likely to be due to the operation of {111}<112> slip systems. Due to the high primary creep strain, the effective stress at the commencement of secondary creep was increased, reducing the creep rupture life. The primary creep strain in the 3Ru alloy was lowered in comparison to 0Ru, as shown in Figure 4c, implying that the inclusion of Ru had an influence on the primary creep strain. Many stacking defects in the γ′ phase and dislocations in the matrix phase were found in the 0Ru alloy. As seen in Figure 8a,b, {111}<112> slip systems operate smoothly in the 0Ru alloy, owing to this observation and the substantial primary creep strain. In the Ru-containing alloy, on the other hand, there were fewer stacking faults in γ′ but a significant number of stacking faults in the γ phase, as compared to 0Ru. Other Ru-carrying alloys exhibit the same phenomena. In Ni-based single-crystal superalloys, Ru is thought to minimize the stacking fault energy [28,29,30]. It was discovered that the dislocation traveled down the γ phase channel in both experimental alloys, causing cross-slip. The shearing of dislocation on the γ-γ′ phases along the [111]<112> slip system, as shown in Figure 8c, necessitates the formation of a stacking fault ribbon from the a/2<110> in the matrix and the propagation of the stacking fault [31]. During the incubation period, the a/2<110> dislocation motion in the matrix channel leaves interfacial dislocations around cuboidal precipitates, forming homogeneous γ/γ′ interfacial dislocation networks. Figure 9 is the TEM micrographs showing the deformation microstructural configuration of both alloys after creep at 850 °C/620 MPa. The dominant mode for plastic deformation under higher applied stress is known to be by the shearing of the γ′ precipitates by dislocations as well as stacking faults and anti-phase boundaries (APBs) [32]. The presence of stacking faults in the γ′ precipitates is observed in both the 0Ru and 3Ru alloys. There is the presence of a stacking fault loop in the Ru-bearing alloy. It can be observed that a considerable number of tangled dislocations are piled up in the γ matrix channel, and plenty of isolated stacking faults, as well as APBs, shear into the γ′ precipitates, but no continuous stacking fault is present in the microstructure.

The dislocation density in the channel of the 0Ru alloy is higher than in the 3Ru alloy. This is one of the main reasons behind the 0Ru alloy’s increased strain rate. Although a fraction of dislocations cutting through γ′ precipitations can minimize dislocation movement resistance, dislocation glide forms a large number of parallel dislocation lines in channels that restrict dislocation reaction between distinct slip systems. The primary creep mechanisms for both experimental alloys are shearing of γ′ precipitates by stacking faults and APBs, as seen in Figure 9b,d. Shearing of γ′ precipitates by dislocation through the process of dislocation gliding in the matrix channel is the major deformation mechanism.

The matrix faults often run across the entire matrix within the {111} plane, i.e., two a/6<112> dislocations are positioned on both sides of the vertical matrix channel on the γ/γ′ interface [32,33,34]. The interface mismatch stress of the two γ/γ′ phases is thought to be the driving mechanism for a/2<011> interface dislocations to become extended dislocations [33].

The TEM dislocation configuration of the experimental alloys is shown in Figure 10 after creep at 1050 °C/270 MPa. It can be seen that dislocations of different configurations cut into the γ′-precipitate phase. The number of dislocations present in the γ′ precipitated phase of 0Ru alloy is more than that of 3Ru alloy. It is considered that the a<101> super dislocation, which cuts into the γ′-precipitate phase, is the main deformation mechanism in this condition of creep [34]. The creep strain rate of 0Ru alloys is slightly higher than that of 3Ru alloys, which may be mainly due to the cutting of a <101> super dislocations into the γ′ phase.

One of the reasons for the long creep life of 3Ru alloys is the dense dislocation grid, which is more prevalent in 3Ru. This dense dislocation grid effectively prevents matrix dislocations from cutting into the γ′ precipitation phase at this high-temperature and low-stress creep condition. When the interface dislocation grids of 0Ru and 3Ru alloys are compared, it is discovered that the dislocation interface grid after creep fracture is not as regular as the interface dislocation grid in the steady-state creep phase. The interface dislocation grid has been continuously evolving during the creep process, and at the third stage of creep, the interface dislocation grid has begun to be destroyed till the specimen breaks. There are numerous dislocations cut into the rafted γ′ precipitated phase, as seen in Figure 10c,d. The building of creep strain in the matrix phase is primarily responsible for this shear process. The a<010> super dislocation decomposes into two a/2<011> dislocations, as seen in Figure 10. This sort of super dislocation has a dislocation-like appearance, and its configuration is mostly due to the limited width between the two a/2<011> dislocations. An a<010> super dislocation must produce a dense dislocation core in which the two a/2<011> matrix dislocations must approach and react with each other in the matrix for shearing to occur in the γ′-precipitate phase without generating a reciprocal domain boundary [35]. When two a/2<011> matrix dislocations with differing burger vectors unite at the γ′ interface to produce a dense a<010> super dislocation, which is then sliced into the γ′-precipitate phase, the a<010> super dislocation decomposes.

The addition of the Ru element not only enhances the stability of the alloy but the solid solution strengthening ability. Generally, the lattice misfit is negative in superalloy due to the smaller lattice parameters of the precipitated γ′ phase than the γ phase. This difference is further enlarged by the Ru-doped. Under high temperature and low stress creep condition, a large negative lattice misfit leads to the increase in interfacial stress and promotes the interaction of dislocations of different slip systems to form a high-density dislocation grid at the γ/γ′ interface. The higher density of the γ/γ′ interface dislocation grid could facilitate the release of most interface misfits. Therefore, the higher degree of lattice misfit can format a higher-density dislocation grid, and the high-temperature creep rupture life will be increased. High-density dislocation grid is a typical deformation mechanism in single-crystal superalloy during high-temperature and low-stress creep, which is verified by many creep experiments [36,37]. Due to the absence of Ru in the 0Ru alloy, it is difficult to form a density dislocation grid with a significant protective effect, resulting in an inferior creep performance than the 3Ru alloy. Therefore, the creep deformation mechanism under high temperature and medium stress is dominated by the high-density dislocation grid, which hinders the active dislocation climbing and cross-slipping at high temperature, and further restrains the decomposition of a<101> super dislocation into two a/2<101> super partials.

The a/2<101> dislocations in the γ phase move together and are entangled at the initial creep deformation under medium temperature and high stress of the 0R alloy. Due to the effort of Ru solute atoms, the critical resolved shear stress that activated at the a/2<101> dislocation in the γ phase could be raised, leading to a longer creep incubation period than the 0Ru alloy. The creep incubation period is a typical characteristic of the creep curve of single-crystal superalloys under medium-temperature and high-stress conditions [38,39]. The Ru element can promote dislocation decomposition when it reaches the decomposition condition because Ru reduces the stacking fault energy of the alloy. Extended dislocation is difficult to recombine at a point before cross-slipping onto a different slip plane, so it can effectively hinder cross-slip. In addition, it is extremely difficult for dislocation climbing to occur at room and medium temperatures. Instead, only dislocation slipping that leads to a longer creep incubation period than the 0Ru alloy in the initial primary creep stage occurs. This is the main difference in deformation mechanisms between the 0Ru and 3Ru alloy in the primary stage of creep at medium temperature and high stress. With the further deformation of the two alloys, a large number of dislocations clustered and entangled at γ/γ′ interface. Dislocation a/2<101> decomposed at the γ/γ′ interface, forming Shockley partial dislocation a/3<112> and a/6<112>. When the stress continues to accumulate, a/3<112> partial dislocation cutting into the precipitated γ′ phase, and a/6<112> partial dislocation remained at the γ/γ′ interface. This period, the deformation mechanism of 3Ru alloy is basically the same as 0R alloy. However, it can be seen in Figure 9 that the stacking fault of 3Ru alloy could hardly pass through the γ′ phase, but it can easily pass through in 0R alloy. It can be inferred that the dislocation first shears into the 0Ru rather than 3Ru alloy under the same temperature and stress conditions.

## 4. Conclusions

The effects of adding 3 wt.% Ru to the Ni-based single-crystal superalloys compared with 0Ru are studied in the present work, and the main conclusions are as follows:Though the alloy including 3 wt.% Ru still keeps a typical dendritic structure in the [001] direction, Ru alters the distribution coefficient, which is called the reverse partitioning behavior, leading to an increase in the eutectic structure of the alloy and decreases in the primary dendrite arm spacing. The average size of the γ′ phase in the 3Ru alloy is a little bit less than in the 0Ru alloy, and the γ′ precipitates are more uniform.Compared with the 0Ru alloy, the 3Ru alloy has a lower creep rate and significantly increased creep life. With the increase in the creep temperature and stress, the creep incubation period of 3Ru alloy becomes longer before the beginning of the primary creep stage. When the creep temperature or stress decreases to a certain value, the incubation period disappears.At medium-temperature and high-stress creep conditions, the addition of Ru effectively inhibits the smooth operation of the dislocation slip system {111}<112> and leads to a low primary creep strain rate of the 3Ru alloy. At high-temperature and low-stress creep conditions, however, the major deformation mechanism changed by forming the dense dislocation grid triggered by the a<101> super dislocation slip, which leads to the high creep property of the 3Ru alloy.

## Figures and Tables

**Figure 1 materials-16-02732-f001:**
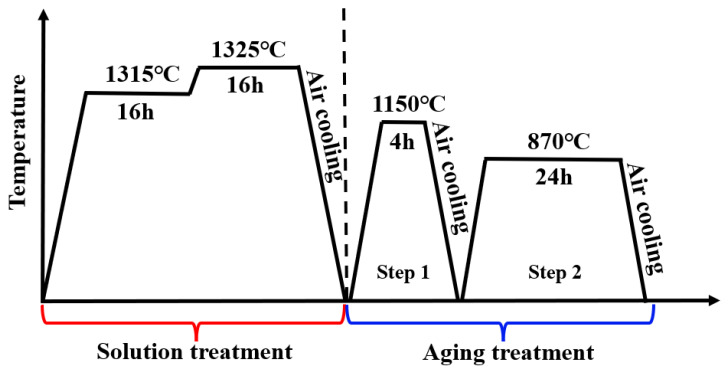
Heat treatment process of the two experimental 0Ru and 3Ru alloys.

**Figure 2 materials-16-02732-f002:**
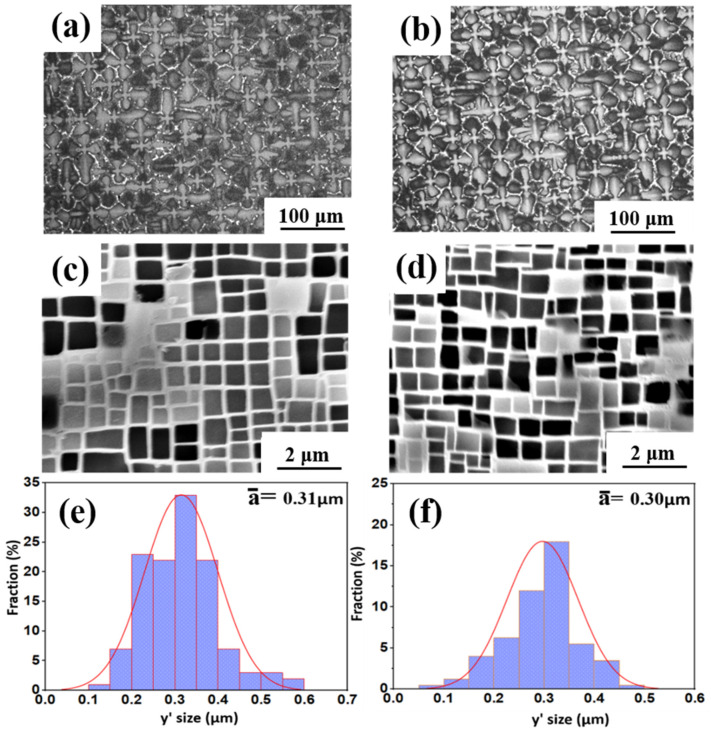
Microstructures of as-cast, heat-treated alloys and average size of γ′ phase, respectively: (**a**,**c**,**e**) 0Ru; (**b**,**d**,**f**) 3Ru.

**Figure 3 materials-16-02732-f003:**
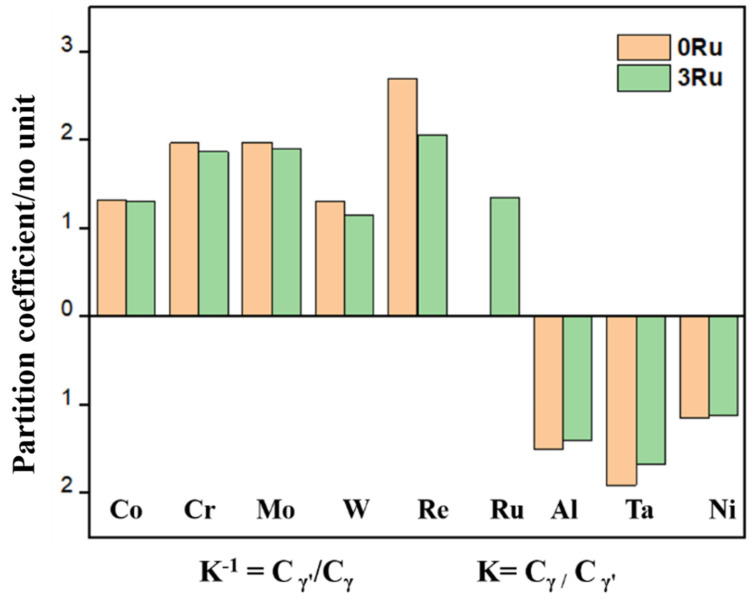
Partitioning behavior of alloying elements in the 0Ru and 3Ru alloys determined by electron probe microanalysis (EPMA). K^−1^ = C_γ′_/C_γ_ and K = C_γ_/C_γ′_, defined as the ratio of the elements in the γ phase to the γ′ phase and the γ′ phase to the γ phase in wt.%, respectively.

**Figure 4 materials-16-02732-f004:**
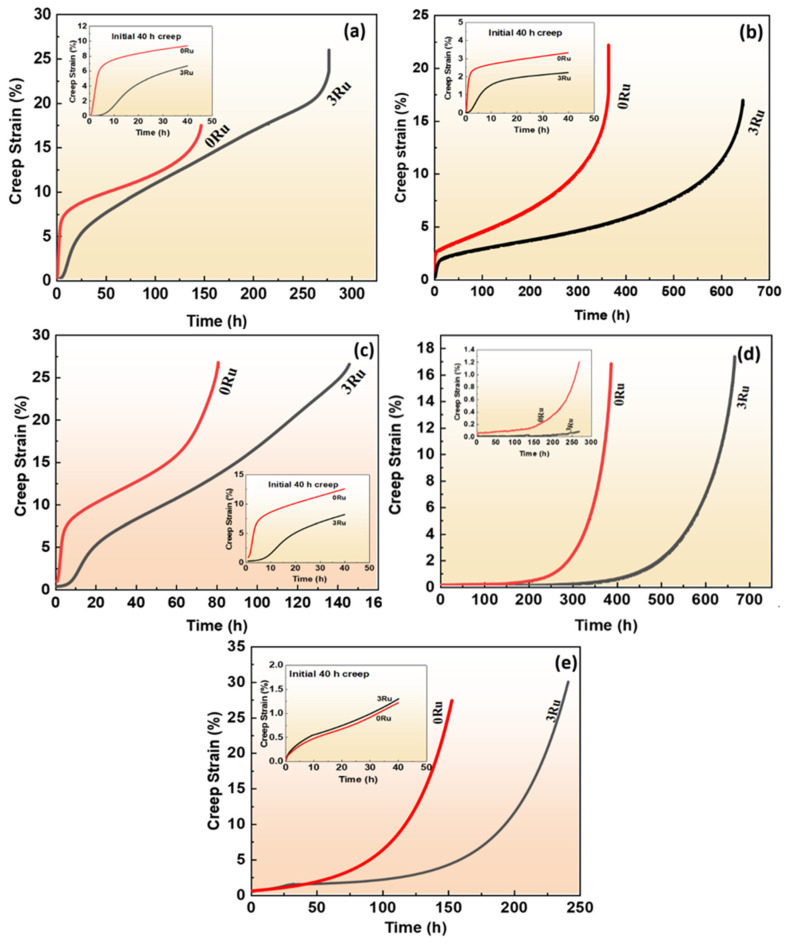
Creep curves of the two experimental single-crystal superalloys at different temperatures and stresses: (**a**) 760 °C/810 MPa, (**b**) 850 °C/620 MPa, (**c**) 850 °C/750 MPa, (**d**) 950 °C/320 MPa, and (**e**) 1050 °C/270 MPa.

**Figure 5 materials-16-02732-f005:**
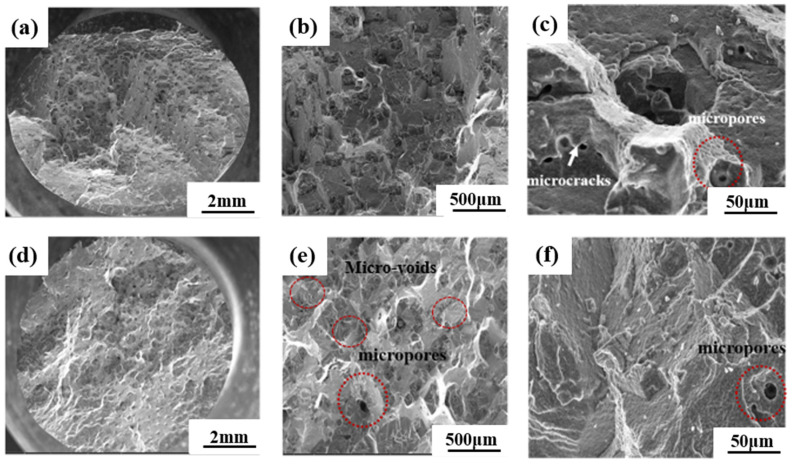
Typical SEM images of creep fracture surface morphology of the two experimental alloys: (**a**–**c**) 0Ru alloy, (**d**–**f**) 3Ru alloy.

**Figure 6 materials-16-02732-f006:**
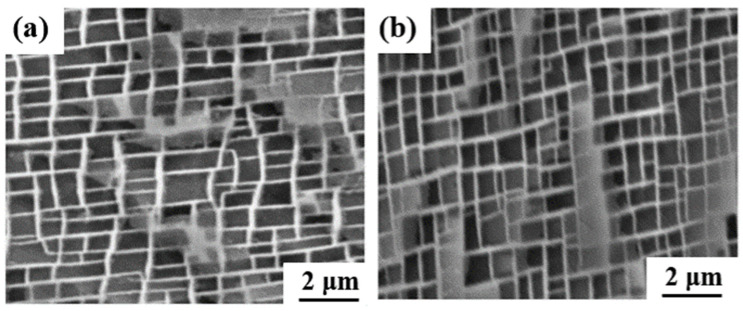
SEM micrographs of experimental alloys after creep rupture: (**a**) 0Ru, (**b**) 3Ru.

**Figure 7 materials-16-02732-f007:**
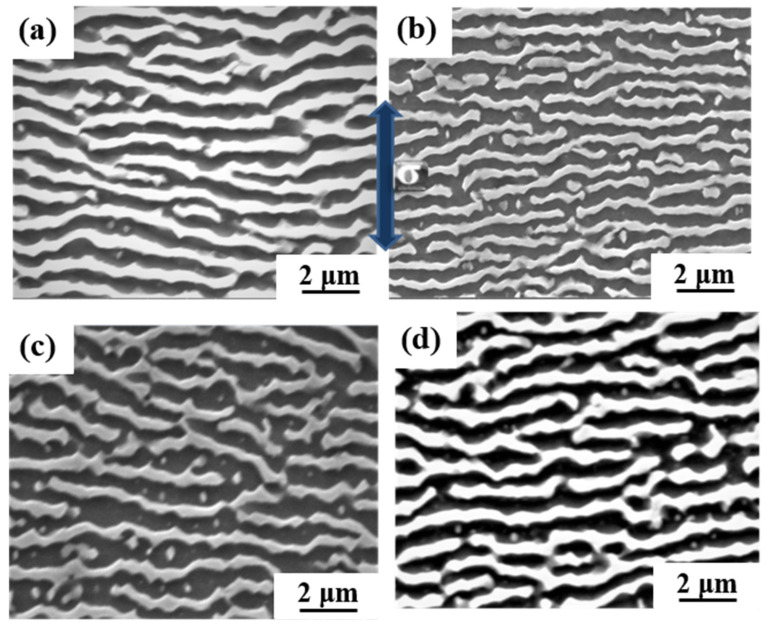
SEM micrographs of experimental alloys after creep rupture at 1050 °C/270 MPa and 950 °C/320 MPa: (**a**,**c**) 0Ru; (**b**,**d**) 3Ru.

**Figure 8 materials-16-02732-f008:**
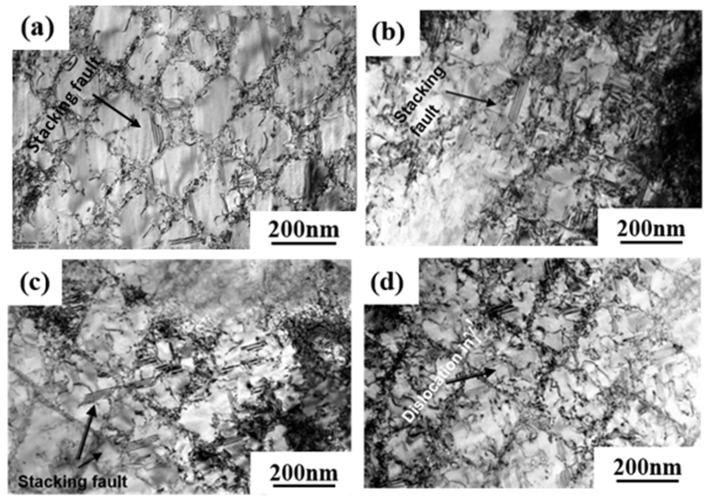
TEM images showing the microstructural configuration of both alloys after creep at 760 °C/810 MPa: (**a**,**b**) 0Ru; (**c**,**d**) 3Ru.

**Figure 9 materials-16-02732-f009:**
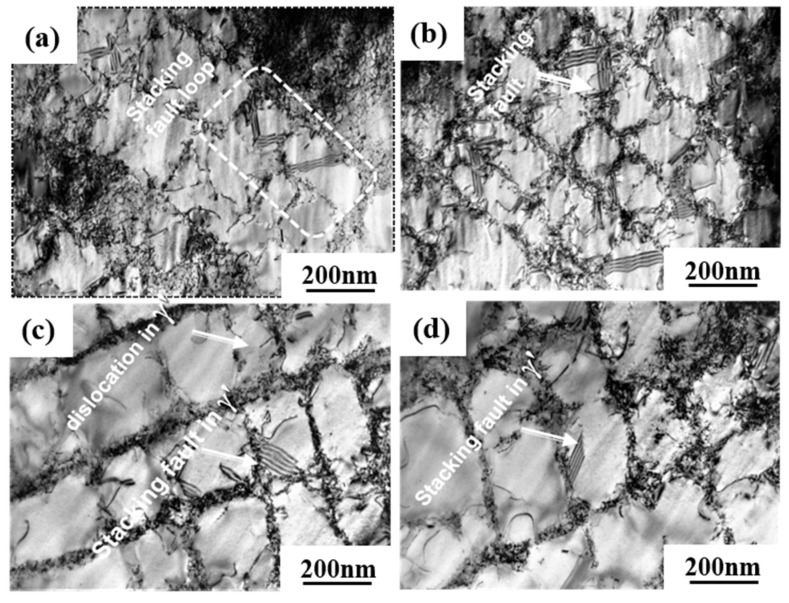
TEM images showing the microstructural configuration of both alloys after creep at 850 °C/620 MPa: (**a**,**b**) 0Ru; (**c**,**d**) 3Ru.

**Figure 10 materials-16-02732-f010:**
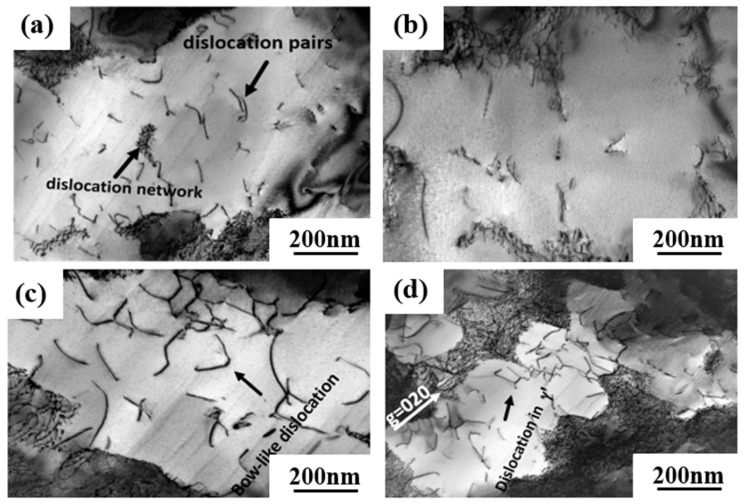
TEM images showing the microstructural configuration of both alloys after creep at 1050 °C/270 MPa: (**a**,**b**) 3Ru; (**c**,**d**) 0Ru.

**Table 1 materials-16-02732-t001:** Nominal chemical compositions (wt.%) of experimental superalloys.

Alloy	Co	Al	Cr + Mo + W + Ta	Re	Ru	Ni
0Ru	12	6	19.4	5.4	0	Bal.
3Ru	12	6	19.4	5.4	3	Bal.

## Data Availability

The data used to support the findings of this study are included within the article.

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
