# Peer review of "Effect of Ru on Deformation Mechanism and Microstructure Evolution of Single-Crystal Superalloys under Medium-Temperature and High-Stress Creep"

_materials, 2023, doi:10.3390/ma16072732_

Round 1
Reviewer 1 Report
Deformation Mechanism and Microstructural Evolution during Medium Temperature High Stress Creep in Ru-free and Ru containing Single-Crystal Superalloys
· English error should be corrected. For example:’…to produced..’ in the abstract.
· Please explain strain incubation in the abstract.
· Abstract and introduction:’…at medium temperature and high stress creep..’: please mention the temperature and the stress range. The objective of the study must be strong.
· Please insert a schematic diagram of heat treatments.
· Figure 2 should be better. A Table can be used for listing partition coefficient.
· “Figure 3d depicts the creep curves of 0Ru and 3Ru alloys at various temperatures and 168 stress levels (900oC/320 MPa).”:incorrect.
· Figure 3, Figure 4, Figure 5 and Figure 6 should be correlated.
· The difference in deformation mechanism between Ru-free and Ru containing single crystal superalloys can be illustrated and discussed well.
Reviewer 2 Report
Reviewer’s Comments:
The manuscript “Deformation Mechanism and Microstructural Evolution during Medium Temperature High Stress Creep in Ru-free and Ru-containing Single-Crystal Superalloys” is a very interesting work. In this work, the effect of Ru element on γ' phase evolution and the deformation mechanism in the fourth generation Ni-based single crystal superalloy was investigated. Results show that Ru element alters the distribution coefficient of other elements in the alloy to produced reverse partitioning behavior, which leads to a difference in microstructure between 0Ru and 3Ru. The addition of Ru triggered strain incubation before the beginning of the primary creep stage, which depends on the creep temperature and stress during creep deformation. While I believe this topic is of great interest to our readers, I think it needs major revision before it is ready for publication. So, I recommend this manuscript for publication with major revisions.
1. In this manuscript, the authors did not explain the importance of the Single-Crystal Superalloys in the introduction part. The authors should explain the importance of Single-Crystal Superalloys.
2) Title: The title of the manuscript is not impressive. It should be modified or rewritten it.
3) Correct the following statement “TEM results revealed that Ru addition inhibits the slip system {111}<112> at medium temperature and high stress creep, which bring considerably low creep rate and high creep life to Ru-containing alloy”.
4) Keywords: The Single-Crystal Superalloys is missing in the keywords. So, modify the keywords.
5) Introduction part is not impressive. The references cited are very old. So, Improve it with some latest literature like 10.3390/molecules27217368, 10.3390/pr10081455
6) The authors should explain the following statement with recent references, “Figure 4 exhibits the typical fracture surfaces morphology of 0Ru and 3Ru experimental superalloy”.
7) Add space between magnitude and unit. For example, in synthesis “21.96g” should be 21.96 g. Make the corrections throughout the manuscript regarding values and units.
8) The author should provide reason about this statement “The number of dislocations present in the γ′ precipitated phase of 0Ru alloy is more than that of 3Ru alloy”.
9. Comparison of the present results with other similar findings in the literature should be discussed in more detail. This is necessary in order to place this work together with other work in the field and to give more credibility to the present results.
10) Conclusion part is very long. Make it brief and improve by adding the results of your studies.
11) There are many grammatic mistakes. Improve the English grammar of the manuscript.
Reviewer 3 Report
This work basically studies the effect of Ru element on γ' phase evolution and the deformation mechanism in Ni-based single crystal superalloy. The authors put a lot of emphasis on the creep mechanism of Ni-based single crystal superalloys by comparing the effect of Ru addition with Ru-free alloys under different temperature and stress creep conditions.
It is clear from the manuscript that a lot of work was invested in this research, however my opinion is that the real novelty of the research is missing comparing with the literature. Several paper can be found, what already deal with the creep mechanisms of the Ni-based single crystal superalloy. Accordingly, I also missed much more relevant literature in the introduction part.
Apart from that, the manuscript contains several grammatical and spelling errors, which make the manuscript very unprofessional. Just a few non-exhaustive examples:
The use of abbreviations is not consequent.
Line 47 and 50: The abbreviations are not explained in the correct place e.g. single crystal superalloy
Line 94: The of explanation of EMPA is not correct.
Line 98: Title of the paragraph 3. 1. is wrong. What is Superalloy Alloy?
line 60: The nomination of 0wt% and 3wt% Ru content alloys is not consistent throughout the manuscript.
Line 93: Somewhere they introduce abbreviation e.g., air cooling (AC), single crystal (SC), but it is not used later in the text.
In the Figure 2., the axis inscription is missing.
The Fig. 7. a and b show the same images.
According to the quality of the journal of Materials, this manuscript contains significant shortcomings, therefore it cannot be accepted for publication in this form.
Round 2
Reviewer 3 Report
Thanks for the response from the authors! The manuscript underwent significant changes, which resulted in an increase in the scientific level of the work. Therefore, the revised manuscript is accepted in this journal.